# An Automated Method for Artifical Intelligence Assisted Diagnosis of Active Aortitis Using Radiomic Analysis of FDG PET-CT Images

**DOI:** 10.3390/biom13020343

**Published:** 2023-02-09

**Authors:** Lisa M. Duff, Andrew F. Scarsbrook, Nishant Ravikumar, Russell Frood, Gijs D. van Praagh, Sarah L. Mackie, Marc A. Bailey, Jason M. Tarkin, Justin C. Mason, Kornelis S. M. van der Geest, Riemer H. J. A. Slart, Ann W. Morgan, Charalampos Tsoumpas

**Affiliations:** 1School of Medicine, University of Leeds, Leeds LS2 9JT, UK; 2Institute of Medical and Biological Engineering, University of Leeds, Leeds LS2 9JT, UK; 3Department of Radiology, St. James University Hospital, Leeds LS9 7TF, UK; 4Center for Computational Imaging and Simulation Technologies in Biomedicine, University of Leeds, Leeds LS2 9JT, UK; 5Department of Nuclear Medicine and Molecular Imaging, University of Groningen, University Medical Center Groningen, 9713 GZ Groningen, The Netherlands; 6NIHR Leeds Biomedical Research Centre and NIHR Leeds MedTech and In Vitro Diagnostics Co-Operative, Leeds Teaching Hospitals NHS Trust, Leeds LS7 4SA, UK; 7The Leeds Vascular Institute, Leeds General Infirmary, Leeds LS2 9NS, UK; 8Division of Cardiovascular Medicine, University of Cambridge, Cambridge CB2 0QQ, UK; 9National Heart and Lung Institute, Imperial College London, London SW3 6LY, UK; 10Department of Rheumatology and Clinical Immunology, University of Groningen, University Medical Center Groningen, 9713 GZ Groningen, The Netherlands; 11Department of Biomedical Photonic Imaging, Faculty of Science and Technology, University of Twente, 7522 NB Enschede, The Netherlands

**Keywords:** aortitis, radiomics, machine learning, convolutional neural network, positron emission tomography/computed tomography

## Abstract

The aim of this study was to develop and validate an automated pipeline that could assist the diagnosis of active aortitis using radiomic imaging biomarkers derived from [18F]-Fluorodeoxyglucose Positron Emission Tomography-Computed Tomography (FDG PET-CT) images. The aorta was automatically segmented by convolutional neural network (CNN) on FDG PET-CT of aortitis and control patients. The FDG PET-CT dataset was split into training (43 aortitis:21 control), test (12 aortitis:5 control) and validation (24 aortitis:14 control) cohorts. Radiomic features (RF), including SUV metrics, were extracted from the segmented data and harmonized. Three radiomic fingerprints were constructed: A—RFs with high diagnostic utility removing highly correlated RFs; B used principal component analysis (PCA); C—Random Forest intrinsic feature selection. The diagnostic utility was evaluated with accuracy and area under the receiver operating characteristic curve (AUC). Several RFs and Fingerprints had high AUC values (AUC > 0.8), confirmed by balanced accuracy, across training, test and external validation datasets. Good diagnostic performance achieved across several multi-centre datasets suggests that a radiomic pipeline can be generalizable. These findings could be used to build an automated clinical decision tool to facilitate objective and standardized assessment regardless of observer experience.

## 1. Introduction

Aortitis refers to inflammatory conditions affecting the aortic wall that cannot be explained by atherosclerosis alone [1,2,3]. It can be an isolated disorder or observed in association with several diseases, including giant cell arteritis (GCA) and Takayasu arteritis (TAK) [3,4]. However, diagnosis of active aortitis presents challenges as symptoms and blood tests can be nonspecific, and treatment can result in severe side effects meaning informed decisions are required [3,5,6].

[^18^F]-Fluorodeoxyglucose Positron Emission Tomography—Computed Tomography (FDG PET-CT) is frequently used to assess patients with suspected aortitis related to large vessel vasculitis (LVV) as FDG avidity identifies areas of high glycolytic activity in the inflamed vessel wall [3,7,8,9,10]. The imaging is often qualitatively assessed based on consensus imaging guidelines [11,12]. Although grading is conducted by imaging specialists, the visual assessment can be subjective and inconsistent [11,13,14,15]. Some semi-quantitative parameters have been utilised but can be vulnerable to several factors and have limited information [16].

Radiomics is a data mining technique involving extraction of quantitative information from medical images referred to as radiomic features (RF) which may help better understand and stratify disease [15,17,18,19]. Radiomics may be a useful technique for aiding in the diagnosis of active aortitis, but the process needs to be automated to deal with a large quantity of data efficiently and facilitate routine clinical use. In particular, vascular segmentation, if conducted manually, can be very time consuming, and is not always reproducible [19,20]. Fully automated segmentation methods using deep learning (DL) convolutional neural networks (CNN) have become increasingly popular as they are both fast and reproducible [21,22,23].

In previous work, a methodological framework for assisting the diagnosis of active aortitis using radiomic analysis of FDG PET-CT was established [20]. This study utilised a small single center dataset to develop a radiomic method and provide a proof of concept that radiomic analysis could add value to the diagnosis of active aortitis.

In this study, the aim was to continue this work by developing, testing and validating an automated radiomic analysis pipeline to assist the diagnosis of active aortitis. This study progresses from the original publication in two ways: firstly, the method was automated by replacing manual segmentation with a CNN and exploring the effect this change had, and secondly the initial findings were validated with data from multiple centres to determine the generalizability and transferability of the method. The pipeline combines automated segmentation, radiomic analysis and machine learning (ML) with the aim of producing a reproducible and standardized method which could be applied to a clinical decision support tool in the future.

## 2. Materials and Methods

There are four key stages to a radiomic diagnostic model (Figure 1: image acquisition, image processing and segmentation, feature extraction, and classification. Each step has multiple sources of variation that can influence final results making the diagnostic model vulnerable to poor reproducibility [24,25]. To mitigate this, TRIPOD guidelines were used [26]. The protocols followed for each stage are described in the following sections.

### 2.1. Image Acquisition

#### 2.1.1. Patient Selection

Figure 2 demonstrates the distribution of the imaging cohorts—training, test and validation. The data acquired from Leeds Teaching Hospitals NHS Trust were split into training and test (80:20) datasets. The training dataset was used to train ML models including optimization of hyper-parameters, and the test dataset was used to confirm initial findings were generalizable. The validation dataset acquired from external centres and used to determine if model performance was transferable to imaging acquired elsewhere.

##### Training and Testing Dataset

The training and test dataset was procured from Leeds Teaching Hospitals NHS Trust from imaging taken between January 2011 and December 2019. The collated data were then split into the training and test dataset (80:20). The inclusion and exclusion criteria for the training and test datasets are described as follows. Patients undergoing FDG PET-CT with a systemic inflammatory response (pyrexia of unknown origin, high acute phase response, weight loss) or suspected active aortitis were identified retrospectively. The ground truth diagnoses for all patients and controls were confirmed by a consultant rheumatologist with 17 years of experience of vasculitis (co-author AWM) based on clinical assessment, blood tests, biopsies and qualitative assessment of FDG PET-CT scans by a dual certified radiologist and nuclear medicine physician (co-author AFS) with more than 15 years of experience of reporting FDG PET-CT. Exclusion criteria included synchronous metabolically active conditions obscuring or interfering with the aorta, such as malignancy. Patients with known LVV were excluded if they did not have imaging evidence of active aortitis. Control patients were excluded if they had activity in the aorta related to atherosclerosis. For LVV patients who had undergone multiple FDG-PET scans, only the first scan that showed aortitis was selected. This study included a combination of newly-diagnosed patients and patients with relapse. The imaging data for the selected aortitis patients and controls were extracted from the institutional PACS (Picture Archiving and Communication System) and pseudoanonymised.

##### Validation Dataset

To evaluate multi-centre transferability, a validation dataset was formed using data from external institutions. The same inclusion and exclusion crieteria were followed but was conducted at the centre of origin. Data from patients recruited to the UK GCA consortium (REC Ref. 05/Q1108/28) [27] with suspected aortitis, and who had FDG PET-CT scans performed as part of routine clinical care at Alliance Medical Ltd (AML) centres in England, were extracted from the organizational PACS (IntelePACS Version 4, Intelerad Medical Systems). The AML centres included Addenbrookes Hospital, Freeman Hospital, Norfolk and Norwich PET CT Centre, Musgrove Park PET-CT Centre, Derriford Hospital, Bradford Royal Infirmary, Guildford Diagnostic Imaging, Sheffield PET-CT Centre, Poole Hospital and The Royal Liverpool University Hospital. The validation cohort was further supplemented by data from the PITA (PET Imaging of Giant Cell and Takayasu Arteritis) (REC approval: 19/EE/0043 Clinical trials registration: NCT04071691, PMID: 36697134) study in the University of Cambridge and Imperial College London.

#### 2.1.2. Imaging Protocol

FDG PET-CT scans were acquired using a standard protocol [11,28,29]. Images were acquired from the upper thighs to the skull vertex in the supine position. Patients fasted for 6 hours before FDG injection, and scanning was conducted 1 hour after injection. Where possible, patients were not currently being treated with glucocorticoids (GC). Nine scanners from three different manufacturers were used (Table 1). Appendix A describes the acquisition parameters in further detail.

The retrospectively gathered FDG PET-CT imaging was converted from DICOM to Nifti file format including converting the PET component to SUV using Simple ITK and PET DICOM (3D slicer extension from the University of Iowa (www.slicer.org/wiki/Documentation/Nightly/Modules/SUVFactorCalculator (accessed on 1 September 2021)).

### 2.2. Image Processing and Segmentation

The segmentation method built into the overall pipeline was a CNN. A subset of the training and test patient dataset (aortitis *n* = 50, control *n* = 25) was manually segmented and given as input to the CNN in order to provide ground truth data to learn. Each FDG PET-CT scan of these patients was segmented manually using 3D slicer and the entire aorta was delineated (Version 4.10.2 (https://www.slicer.org/ (accessed on 1 April 2020)) [30,31]. The CT component was used as the main reference as it provides more anatomical information, but the result was checked against the PET scan. The Dice similarity coefficient (DSC) (Equation (Equation 1)) [32] was used to evaluate segmentation quality:(1)$$DSC=\frac{2|A⋂B|}{\left|A|+|B\right|}$$

The PET and CT components, and segmented masks were then resampled to a 4 mm isotropic voxel size to ensure uniform sampling across the entire cohort. Linear interpolation in Simple ITK was used for downsampling. This voxel size was selected as it was the lowest resolution of the three scanners in the training and test dataset meaning downsampling alone was applied. A lower resolution was present in multi-centre data collected later (5.47 mm), but the 4 mm voxel size was maintained to ensure a valid comparison, and to keep an integer voxel size preventing rounding errors. The images and masks were also cropped to the same window size (144 × 144 pixels) as the CNN required the same slice sizes. Data were manually checked to ensure that the aorta was central and unaffected by the crop.

A CNN with U-Net architecture was built for automated segmentation (Tensorflow Version 2.4.1). The full architecture is shown in Figure 3. Training was undertaken on ARC4, and part of the high-performance computing facilities at the University of Leeds, UK. On ARC4, a single NVIDIA V100 GPU (graphics processing unit) was used. In total, training and then segmentation of all data took 11:51:20 (HH:MM:SS). The average segmentation time per patient was 1 min 12 s compared to an average of 30 min per patient for manual segmentation.

The manually segmented dataset was split into training and testing cohorts for the development of the CNN(70:30), and each CT image was read in slice by slice with its corresponding labelled slice as the input layer. The performance of the CNN was measured using the DSC. The batch size was set to 32 slices. The number of epochs was set to 100 with early stopping if the loss function (DSC loss) did not improve, which led to training stopping at 41 epochs. The activation function was a leaky rectified linear unit (ReLU). Convolution stride was 1, and pooling stride was 2. Kernel size was 3 × 3 for convolution and 2 × 2 for pooling. Once trained, the entire patient dataset was provided as input, and the predicted segmentations were output. Small ’islands’ were found in the predicted segmentations. These were clusters of pixels in the background of the scan that were several orders of magnitude smaller than the aorta. These were removed by creating new segmentations that only retained the largest cluster of pixels in the slice using Python packages Numpy (Version 1.18.1) and Simple ITK (Version 2.01). The segmented slices were then reassembled into 3D volumes for use in feature extraction (Section 2.3.2).

### 2.3. Feature Extraction

#### 2.3.1. Qualitative Grading of Vessel Wall FDG Activity

All scans were evaluated based on EANM/SNMMI guidelines [11] and assigned a vascular uptake score by an experienced radiologist (supervisor AFS):

0: no uptake (less than mediastinum)

1: low-grade uptake (less than liver)

2: intermediate-grade uptake (equal to liver), (possible aortitis)

3: high-grade uptake (greater than liver), (positive active aortitis)

#### 2.3.2. Feature Extraction

Radiomic features encompass a large number of quantitative parameters. These features range from simple, such as SUV (standardised uptake value) metrics (Equation (Equation 2)), to more complex descriptors of the shape and spatial relationships between individual voxels of imaging data [33,34].

SUV metrics are part of the larger group of radiomic features but will be referred to independently when studied separately. As they are more established in clinical use, their diagnostic utility was explored alone and as part of the larger group. The SUV of a region of interest (ROI) can be averaged (SUV_mean_) or the maximum determined (SUV_max_). However, SUV measurements can be influenced by several factors such as the size of the volume of interest, image noise, concentration of glucose in plasma and body habitus [16]. SUV metrics were used instead of target-to-blood pool ratio as liver is a more common reference point as discussed in Section 2.3.1:(2)$$SUV=\frac{radioactivity\;concentration}{\;injection\;dose\;\left(MBq\right)\;/patient^{\prime }s\;weight\;\left(kg\right)}$$

Radiomic features (*n* = 102) were extracted with Pyradiomics (Version 3.0.1, radiomics. io/pyradiomics). A further five SUV metrics (SUV_x_) were calculated separately using Numpy (Version 1.18.1) and Simple ITK (Version 2.01) and added to the radiomic features dataset. Each SUV metric was calculated as follows:SUV 90th Percentile—90% of the voxel’s SUV value fall below this number;SUV mean—the mean SUV value in the region of interest;SUV maximum—the maximum SUV value in the region of interest;SUV x (x = 50, 60, 70, 80, 90)—mean of the voxels that are equal or greater than x% of SUV maximum.

In both cases, the radiomic features were extracted from the entire segmented 3D volume of the aorta in the PET image [35]. In most cases, Pyradiomics is broadly compliant with the IBSI standards but deviates in some cases as described in their documentation (https://pyradiomics.readthedocs.io/en/latest/faq.html (accessed on 1 November 2022)). This will affect some of the extracted features where they rely on gray value discretization. Features were calculated with a SUV bin width of 0.075. This bin width was determined by dividing the maximum SUV value in the segmented areas across the whole dataset by 64—a commonly used bin number in radiomics. No filters were applied through Pyradiomics, and all other parameters were left as default.

A complete list of all radiomic features and SUV features (*n* = 107) extracted is provided in Table A2.

### 2.4. Diagnosis with Machine Learning Classifiers

#### 2.4.1. Diagnostic Utility of Individual SUV Metrics and Radiomic Features

The diagnostic utility, also referred to as diagnostic performance, of the following methods was measured with AUC primarily, along with balanced accuracy as confirmation. Balanced accuracy was used as it adjusts for imbalanced datasets and allowed for comparison between our training, test and validation datasets. The AUC of the validation dataset was prioritised as it demonstrated both generalizability to other datasets and transferability to other institutions which is vital for clinical use [23]. As the benchmark AUCs for qualitative assessment of PET-CT in suspected aortitis quoted in the literature are 0.81–0.98 [11], any AUC value greater than 0.8 was considered a good performance. Where possible, methods with any balanced accuracy across the three cohorts ≤ 50% were discounted. Cases where AUC was high but accuracy was low occur due to a bias towards the positive diagnosis.

The diagnostic utility of all radiomic features and SUV metrics were first evaluated individually using logistic regression classifiers (Sci-kit Learn Version 0.23.2). While SUV metrics can be included as radiomic features (Section 2.3.2), they were separated and compared to all remaining radiomic features at this stage to determine if the newer radiomic features added value. To train the logistic regression classifiers, the hyper-parameters for each feature were tuned using the Sci-kit Optimise function BayesSearchCV using the training cohort with stratified 5-fold cross validation meaning the ratio of patients to controls in each fold was equal to the ratio in the total cohort. The hyperparameter optimization method was changed to BayesSearchCV from GridSearchCV from the previous study as it more thoroughly searches the parameter options [20]. The final diagnostic model for each individual feature was then trained with the best hyper-parameters on the training cohort with stratified 5-fold cross validation. The trained model was then applied to the test and validation dataset.

#### 2.4.2. Forming Radiomic Fingerprints

Individually radiomic features can be used as metrics, but, when used collectively, they can provide complimentary information to improve diagnostic performance [36]. Using all or most extracted radiomic features can introduce a significant amount of redundant information and creates noise in the diagnostic model [37]. Therefore, radiomic fingerprints were created with the extracted radiomic features. Three radiomic fingerprints were built using the methods described below.

Fingerprint A was produced by selecting features with high individual diagnostic utility based on their training dataset performance in Section 2.4.1: AUC ≥ 0.5, balanced accuracy ≥ 0.5. Features were filtered using Python package Pandas (Version 1.1.4). Highly correlated features were then removed. For every combination of feature pairs, if the correlation coefficient was >0.9, the feature with the lower AUC was removed.

Fingerprint B was formed using principal component analysis (PCA). PCA represents a large set of variables as a smaller set of principal components by finding relationships between features and combining them to reduce redundancy and minimize loss of information. PCA was applied using Sci-kit Learn (Version 0.23.2). The fingerprint was formed with principal components needed to account for at least 90% of variance in the radiomic data.

Fingerprint C used the Sci-kit Learn (Version 0.23.2) random forest ML classifier. The classifier has intrinsic feature selection so all 107 extracted features were provided as input, and the classifier will select the features that produce the best performance.

#### 2.4.3. Diagnostic Utility of Fingerprints

The diagnostic utility of radiomic fingerprints A and B was evaluated using the same methodology described in Section 2.4.1, but additional ML classifiers were tested alongside logistic regression [38,39,40]. Ten different ML classifiers were built, trained and tested (Sci-kit Learn Version 0.23.2): support vector machine, random forest, passive aggressive, logistic regression, k nearest neighbours, perceptron, multi-layered perceptron, decision tree, stochastic gradient descent and gaussian process classification. Stochastic gradient descent refers to the SGDClassifier within Sci-kit learn that uses stochastic gradient descent optimization on several different linear classifiers. The specific linear classifier is determined as part of hyper-parameter tuning.

Fingerprint C was evaluated as in Section 2.4.1 using only the Random Forest Classifier as it uses the embedded feature selection in this ML classifier.

#### 2.4.4. Statistical Analysis

AUC was used as the key metric for determining diagnostic utility of the tested diagnostic models and ranking accordingly. Balanced accuracy used a confirming metric. The confidence intervals were determined using Delong’s test [41]. The final comparison of models was conducted using the *p*-value derived from Delong’s Test.

### 2.5. The Influence of Variation in Method

#### 2.5.1. Harmonization

The effect of the ComBat Harmonization method (neuroCombat, Version 0.2.7) was explored. It is a technique used to reduce the effect of different imaging protocols on radiomic features [42,43,44]. These factors cannot be standardized retrospectively without reducing the size of the dataset, so harmonization is recommended to minimize the effect [44]. The overall dataset (training, test and validation combined) was grouped in batches as shown in Table 1 based on similar imaging protocol parameters. The effect of harmonization was explored in the previous study [20]. No significant improvement was achieved but as there is more variation in data in this study the comparison was repeated. The key results with and without harmonization were compared.

#### 2.5.2. Segmentation

A CNN was utilised as the segmentation method in this study to automate the radiomic workflow. A subset of patients (50 aortitis and 25 controls) had both manual and automatic segmentations. The above methods were repeated on these patients using both segmentations separately in order to compare the effect the segmentation method had on performance. There were insufficient numbers to have a test and validation cohort so only training values were compared.

#### 2.5.3. Imaging Sources

This study used multi-centre data from the UK as a validation cohort. This cohort was varied but generally followed the same diagnosis pathways set out by the National Health Service (NHS), standard of care imaging protocol, and were imaged in the same time period. Later in the study timeline, another dataset was collated from UMCG, Groningen, The Netherlands consisting of 40 GCA patients and 20 controls. Different imaging protocols were followed, and the images were acquired on scanners with higher image resolution—a range of Siemens Biograph scanners using the reconstruction methods PSF + TOF 3i21s most often, and PSF + TOF 4i5s and OSEM3D 3i24s occasionally. These variations are known to highly influence radiomic results [45].

The robustness of our method to highly varied data was tested with and without harmonization. The images were preprocessed and segmented in the same way as the previous cohorts before being included in the validation cohort and the diagnostic utility retested.

## 3. Results

### 3.1. Image Acquisition—Patient Characteristics

Overall, 114 participants were included in the training, test and validation datasets collectively (Table 2). The age of the patients and female predominance reflects the typical demographic of patients with LVV, the most common cause of which is GCA. The sensitivity of FDG PET-CT is significantly reduced within a few days of starting GC treatment, so GC (prednisolone) doses were zero at the time of scanning unless stated otherwise [46]. CRP and ESR are laboratory markers of inflammation used in clinical care.

Less clinical data were available for the validation dataset, but, as shown in Table 2, the gender distribution, LVV Type, prednisolone dose, CRP, ESR, blood glucose and median age of all datasets are similar.

### 3.2. Segmentation

The manually segmented data had a mean DSC of 0.91 when a sample was compared to segmentations conducted by a second observer. The CNN achieved a mean DSC of 0.66 (median 0.72) before small `islands’ were removed and 0.71 (median = 0.80) after when compared to the original manual segmentations used for training. The time taken to segment the aorta automatically per patient was 1 min 12 s.

An example of a CNN segmentation is shown in Figure 4.

### 3.3. Qualitative Grading of Vessel Wall FDG Activity

Recent guidelines advocate qualitative grading of PET-CT scans based on FDG activity in the aortic wall relative to the liver [11]. Table 3 shows the grades assigned to the training, test and validation cohorts respectively by an experienced radiologist on retrospective review of the images.

### 3.4. Diagnostic Utility of Individual SUV Metrics and Radiomic Features

Figure 5 demonstrates the performance of SUV metrics in a logistic regression classifier where higher accuracy and AUC indicate good diagnostic utility. In general, SUV metrics performed poorly when accuracy was considered. The SUV 90th percentile performed better consistently across all three cohorts with a validation AUC of 0.85 and a balanced accuracy of 71%.

The five-best performing radiomic features, when used individually in a logistic regression classifier, are shown in Figure 6. Performance was based on validation AUC but a minimum balanced accuracy of 50% had to be met across the training, testing and validation cohorts. In some cases, a radiomic feature would perform well in either testing or validation AUC but had poor accuracy. The radiomic features given in Figure 6 suggests that heterogeneity is an important characteristic in distinguishing aortitis from controls.

The performance of all individual radiomic features and SUV metrics in logistic regression classifiers, and in all three cohorts, are listed in Table A2.

### 3.5. Diagnostic Utility of Fingerprints

Fingerprint A was based on minimum thresholds of diagnostic performance for each feature and a maximum correlation to other features. Random Forest performed consistently across the training, testing and validation cohorts in both AUC and balanced accuracy (Figure 7), suggesting that this method may have multi-centre transferability. The performance of all explored ML classifiers is shown in Appendix A
Table A3.

Fingerprint B was based on PCA. For this fingerprint, the best validation AUC was achieved by a neural network classifier with a validation AUC = 0.90 (Figure 8). The performance of all explored ML classifiers is shown in Appendix A
Table A4. Appendix Table A4 also shows that, despite the neural network being the best performing classifier by validation AUC, the ranking metric used in this study, several other classifiers performed similarly and more consistently across datasets and with smaller confidence intervals.

Fingerprint C used the feature selection that is intrinsically part of Random Forest classification and did not include any other ML classifiers. This method produced good results in both the testing and validation cohorts, demonstrating that Fingerprint C is a promising method for the diagnosis of aortitis. Figure 9 displays the ROC curves for Fingerprint C. As this method only used Random Forest, it was not tested in all ML classifiers.

### 3.6. Comparison of Selected Features

Table 4 shows the top 10 features (by Validation AUC and feature importance metric respectively) selected in Fingerprint A and Fingerprint C. As Fingerprint B used PCA and produces new components, it is not simple to directly compare them. Table 4 demonstrates that heterogeneity is important in distinguishing aortitis from controls. This is confirmed by earlier results in Section 3.4.

### 3.7. Summary of Key Results

Table 5 summarizes the best results from each of the explored methods for diagnosis of aortitis. The best result was determined as described in each of the previous sections, but, in all cases, validation AUC was used to initially rank the results and where possible results with a balanced accuracy ≤50% were removed. While the displayed results ranked the best in each method by the given criteria, none were significantly better than each other with respect to AUC (*p*> 0.05, DeLong’s Algorithm [41,47]).

### 3.8. Influence of Variations in Method

The results given in Table 6 are from the same diagnostic models evaluated in Table 5 but with harmonized data as input instead. This demonstrates that feature harmonization has little influence on diagnostic utility in this scenario.

The results given in Table 7 compare the training AUC values for manual and CNN segmentation. These results show comparable performance for both segmentation methods.

The results given in Table 8 demonstrate which methods are affected by using a heterogeneous data set. The heterogeneity mostly involved larger variations in image acquisition protocol. Only validation data are shown as the new dataset was added to the validation cohort. CNN segmentations on this data achieved a median DSC of 0.71 when compared to manual segmentations. This demonstrates that standardization of imaging acquisition is required for most of the explored radiomic based diagnosis methods. Fingerprint A demonstrates the most robustness, albeit a small decrease in diagnostic utility.

## 4. Discussion

This study presents an automated pipeline to assist diagnosis of active aortitis using CNN segmentation, radiomic analysis, and ML classifiers. Key results are summarised in Table 5. The different diagnostic models performed well and similarly to both each other and the standard of care qualitative assessment. Using radiomic fingerprints had the advantage of reducing the size of the confidence intervals. Fingerprint A demonstrated the most robustness.

### 4.1. Segmentation Automation

The main component in automation was aortic segmentation using a CNN which achieved a median DSC of 0.80 and allowed the diagnostic models to achieve a good performance. A good diagnostic performance or utility was defined as a Validation AUC ≥ 0.8, and is therefore similar to the benchmark AUCs for qualitative assessment of PET-CT in suspected aortitis [11], and a (balanced) accuracy > 0.5 in all three cohorts. When compared to the performance using manual segmentations (Table 7), comparable results were achieved. Automating the method reduces the likelihood of inter and intra observation, increasing reproducibility [48]. It also makes routine clinical adoption a more realistic proposition.

### 4.2. Multi-Centre Transferability

Sollini et al. concluded in their systematic review that the lack of external validation was the key issue preventing radiomics translating into routine clinical practice [49]. Some multi-centre diagnostic performance was shown in the proposed methods. SUV metrics performed well in training cohorts but did not demonstrate good transferability to the testing or validation cohort from other institutions. SUV 90th Percentile demonstrated the most diagnostic utility from all the explored SUV metrics and did so in all three cohorts. In future work, it may be worth investigating the effect of adjusting for lean body mass rather than body weight, as is the case for SUV metrics, as the results from van Praagh et al. suggest that this could be more reliable [16]. Several individual radiomic features produced high AUC values and met the minimum accuracy values. In particular, features based on heterogeneity performed well across all three cohorts with the highest validation AUC coming from GLDM Dependence Entropy (AUC = 0.91) and first order Energy achieving the best compromise between AUC and accuracy.The features selected in Fingerprint A and C further demonstrate that heterogeneity is important in distinguishing aortitis (Table 4).

Fingerprint A was formed with high performing individual features with highly correlated features removed. It performed well with Random Forest in all three cohorts achieving AUC and accuracy values above the minimum thresholds stated earlier. This fingerprint also displayed the most robustness to using a heterogeneous imaging protocol in Table 8. This demonstrates good generalizability and transferability which are important for clinical use [49]. Fingerprint C (Random Forest—all features) performed similarly but was detrimentally affected by the changes in imaging protocol. This suggests it is beneficial to still pre-select features before using random forest.

### 4.3. Limitations

It would be preferable to only analyse the aortic wall, but a segmentation method which reliably distinguishes wall from the lumen in non-contrast enhanced CT has not been developed to the extent that it reliably worked on the patient cohort in this study. A segmentation method that achieves this with thresholding can be applied to PET scans with high aortic wall activity, but it would not identify non-inflamed aortic wall in the control dataset so this segmentation method would not be feasible for diagnostic purposes. Future studies that used only aortitis cases with radiomic analysis might be able to overcome this issue using thresholding: for example, prognostic/stratification studies. Location of inflammation in the aortic wall could also be considered for differentiating causes of aortitis as this varies. These were beyond the scope of the present work. Most other aortic wall segmentation methods require contrast enhancement. One method that does not use contrast enhancement was developed by Piri et al. [50]. They utilised a CNN to produce whole aorta segmentations and then labelled the wall as a predefined thickness inside and outside of the edge, giving a 5 mm wall thickness in total. This gives a good approximation of the aortic wall but is not precise, does not account for inter and intra patient variation, and is not flexible to deal with anatomical alterations caused by aortitis. While a 5 mm thickness is achievable on CT, it would result in a one pixel thick wall on PET used in this study (voxel size 4 × 4 × 4 mm) making any second or higher order features limited. Future work could explore whether the trade off between limiting potential features or including the lumen is worthwhile.

Another limitation is that patients with atherosclerosis were excluded from the study cohort, for both aortitis patients and controls, as atherosclerosis can also lead to FDG uptake in the vessel wall [51,52,53]. This was part of the exclusion criteria as the purpose of the study was to initially develop an artificial intelligence-based pipeline using unequivocal cases and controls. The next stage would be to test this pipeline on the whole spectrum of those presenting with suspected aortitis; atherosclerosis is common in this age group [12]. Another group has reported promising results using SUV metrics [54]. Similarly, it would be of interest to determine whether any of the radiomic features with high diagnostic utility can detect aortitis after treatment has started as this currently limits the use of PET imaging. FDG PET-CT is mostly used for baseline imaging of aortitis for diagnosis as GCs reduce its sensitivity [46,55,56]. While the diagnostic accuracy decreases significantly after 10 days, uptake is not eradicated completely. Van der Geest et al. determined that FDG PET still had some moderate diagnostic utility for monitoring treatment but that the individual results were highly variable and any conclusions drawn from imaging should only be interpreted in the context of clinical presentation [57]. This evaluation was based on visual assessment which is based on vessel activity compared to the liver and the distribution of uptake. Potentially, some of the radiomic features that demonstrated a high diagnostic performance, but are based on information that is not easily appreciated by eye, could help utilise FDG PET for monitoring aortitis.

Several components of radiomics can be replaced by DL methods adding many benefits such as automation and reproducibility. However, these methods require large datasets that are not always conceivable. They also need to be interpretable and understandable in a clinical context to encourage trust, avoid unnoticed bias in training data, and overcome privacy, legal and accountability issues [23,37,58]. These limitations do not eliminate the use of DL and are likely to be easier to overcome in coming years. It does leave room for other techniques like handcrafted radiomic features and simpler ML classifiers. While DL is popular, its application to steps in the radiomic workflow does not always produce better results than other well-established methods.

### 4.4. Harmonization and Standardization

There is still some debate as to the validity of harmonisation with ComBat [19,59,60]. Orlhac et al. stated that ComBat is only appropriate in situations described in their guide [61]. Papadimitroulas et al. described several other alternatives to ComBat but also concluded that ComBat performed well overall [23]. As this study uses data from several institutions and scanners, harmonization was explored and showed no significant difference in diagnostic utility. A key disadvantage to using ComBat for harmonization is that scans that do not belong to a predefined batch cannot be harmonized. In order to define a batch, a minimum of 20 samples are required, which was not achieved in some of our batches and is not feasible in many smaller centers.

A key point discussed in numerous radiomic studies and reviews is the need for standardized methodology. Standardization of imaging protocols was not feasible as this was a retrospective study with insufficient data to exclude patients based on imaging protocol. All steps after reconstruction and before feature extraction were kept consistent as this has been proven to have a significant effect [62]. Table 8 demonstrates the effect of using input data from centres following different imaging protocols. Fingerprint A showed the most robustness, making it promising for transferability. TRIPOD reporting guidelines were adhered to in this project to ensure transparency of methodological details [26]. Feature extraction software (PyRadiomics) that mostly adheres to IBSI radiomic feature standardization was utilized. The IBSI definitions are discussed in their paper by Zwannenburg et al. [34]. Deviations from these definitions are discussed in the user documentation (https://pyradiomics.readthedocs.io/en/latest/features (accessed on 1 November 2022)) and accompanying publication [35]. While standardization is important, specific recommendations for steps are rarely made as optimal methods vary based on modality, condition and application. Some specific recommendations are published for PET imaging in LVV, but there are little or no studies reporting use of radiomics in this setting, meaning there is no specific guidance. The results of this project provide initial results but further optimization of each step could be explored to produce specific advice to this application. Meanwhile, thorough reporting of methods is sufficient to overcome most issues caused by a lack of standardization. As IBSI found, even with well-defined rules, there can be discrepancies in application, so following guidelines such as IBSI, STARD or TRIPOD when reporting can help convey the most important details [26,34]. Some decisions made in this project have been in areas still debated in literature. Examples include how to define the bin width or bin number when conducting gray level discretization, or whether to upsample or downsample when spatially resampling. While they are not limitations, they may be improved upon in future studies [22,34].

## 5. Conclusions

The purpose of this study was to develop and validate an automated pipeline that assists the diagnosis of active aortitis. The pipeline included an automated segmentation method with a CNN, radiomic analysis and ML.

The different diagnostic models performed well and similarly. Fingerprint A demonstrated the most robustness and was built using the best performing individual features whilst removing correlated features.

These findings demonstrate a radiomic pipeline can be generalizable and transferable. They could be used to build an automated clinical decision tool which would facilitate objective and standardized assessment regardless of observer experience.

## Figures and Tables

**Figure 1 biomolecules-13-00343-f001:**
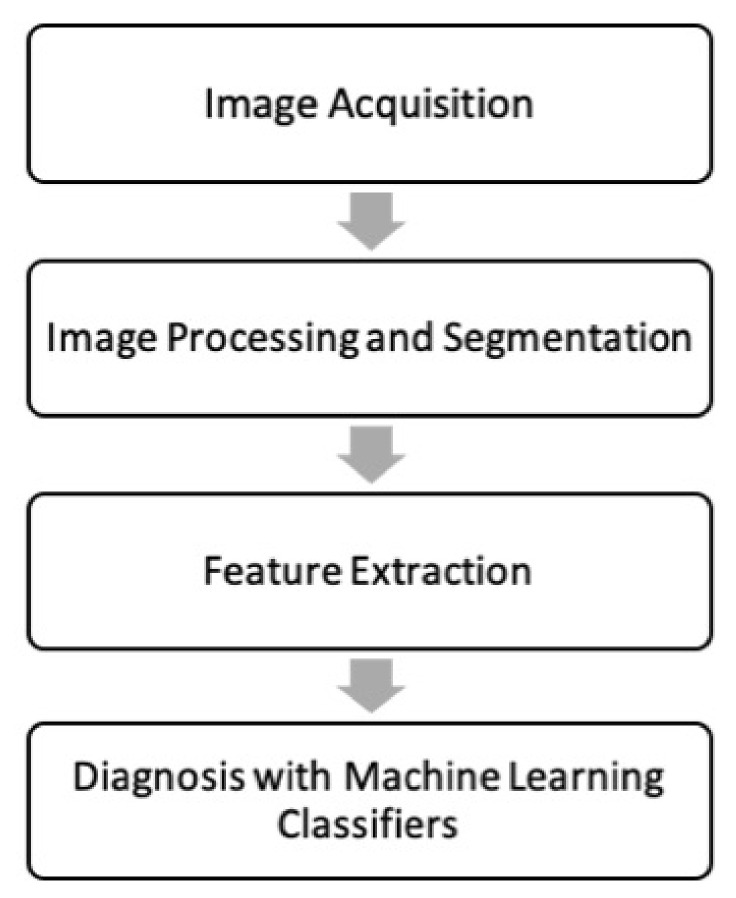
The steps of a radiomic diagnostic model.

**Figure 2 biomolecules-13-00343-f002:**
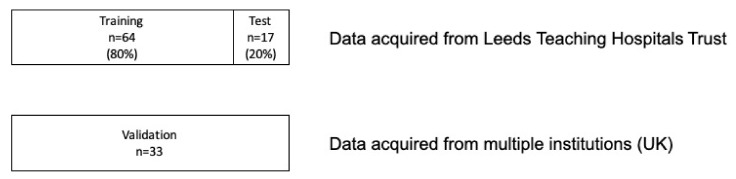
The distribution of datasets into training, test and validation cohorts.

**Figure 3 biomolecules-13-00343-f003:**
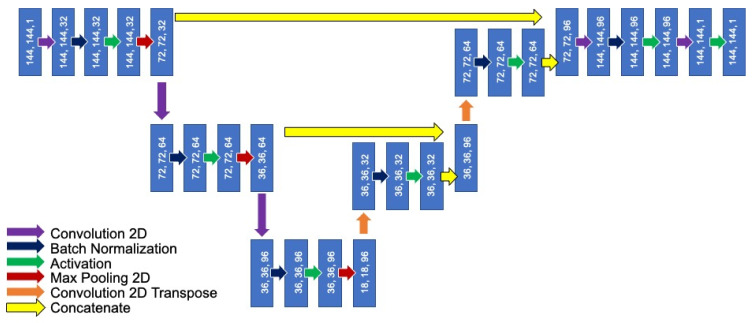
Architecture of convolutional neural network (CNN) used to segment the aorta.

**Figure 4 biomolecules-13-00343-f004:**
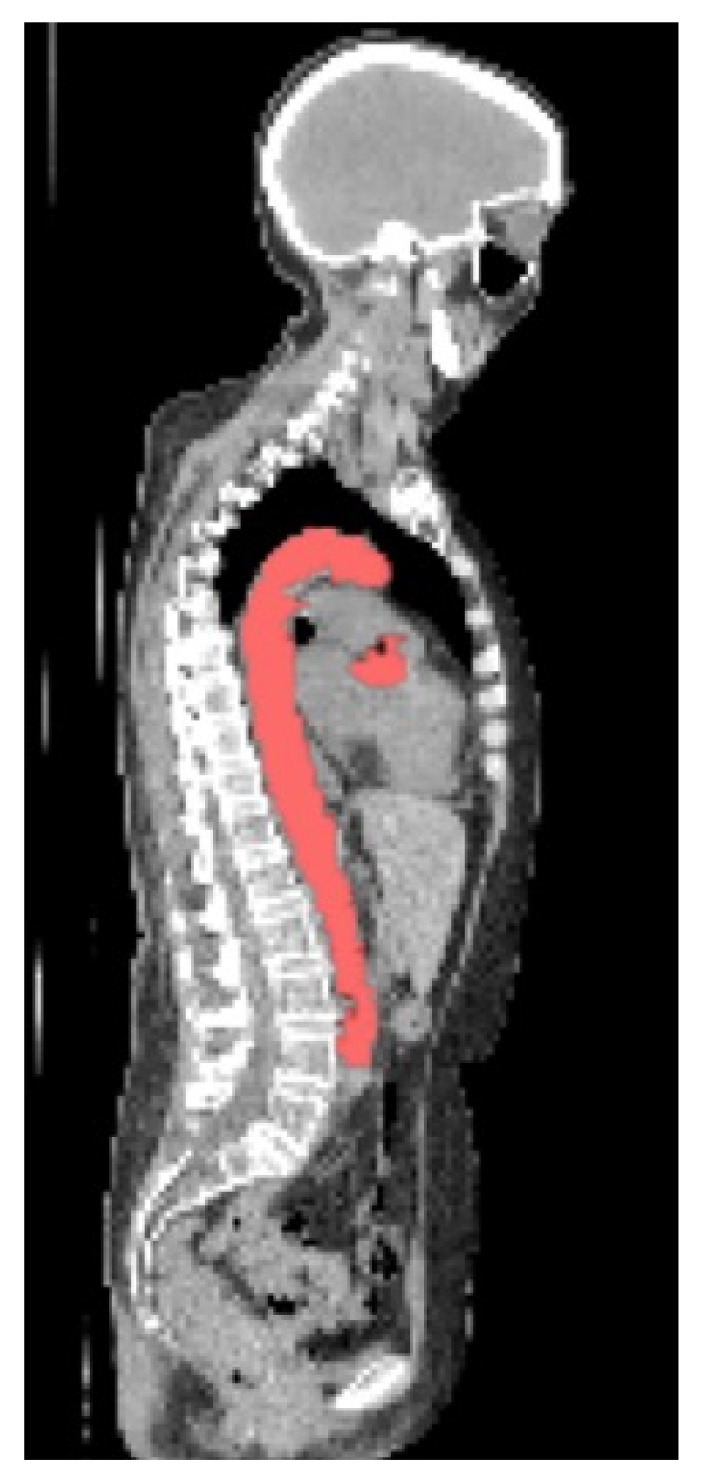
Segmentation from CNN with small `islands’ removed.

**Figure 5 biomolecules-13-00343-f005:**
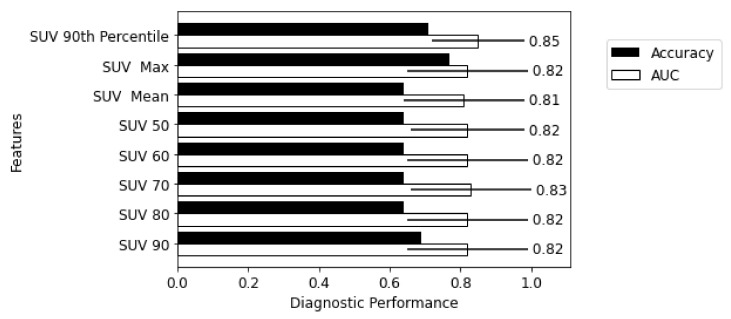
Diagnostic utility of individual SUV metrics.

**Figure 6 biomolecules-13-00343-f006:**
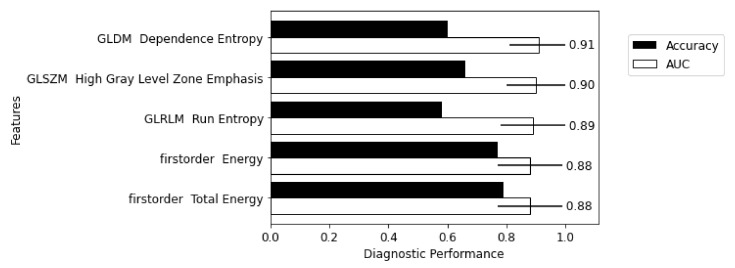
Diagnostic utility of the five highest performing individual radiomic features—performance ranked by validation AUC with a balanced accuracy above 50%.

**Figure 7 biomolecules-13-00343-f007:**
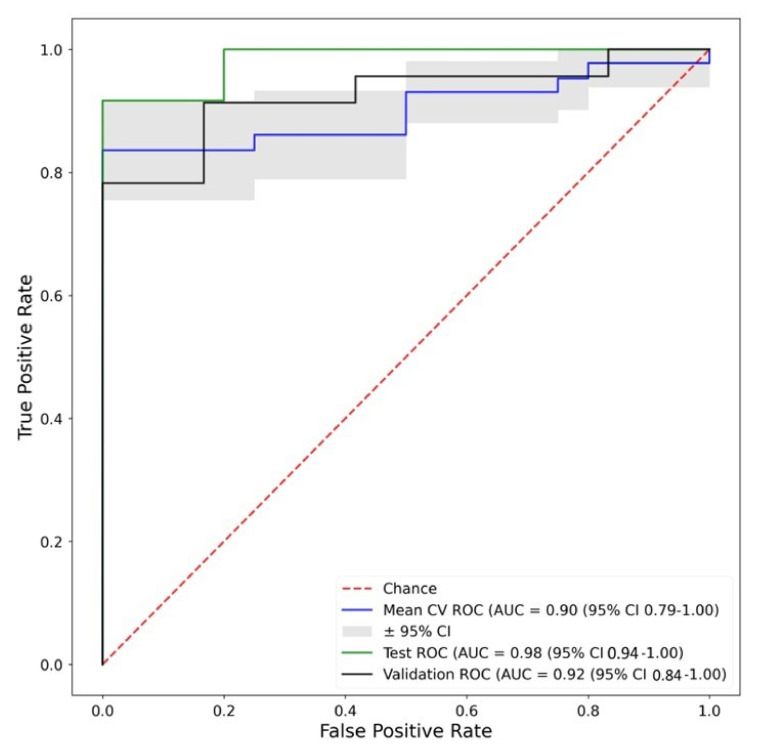
ROC curves of the best performing (by validation AUC and minimum accuracies) machine learning classifier trained on Fingerprint A—Random Forest. Corresponding Accuracies—Training: 0.74, Test: 0.8, Validation: 0.75. Key: Mean CV ROC—Mean cross validation ROC from training dataset, Test ROC—ROC from test dataset, Validation ROC—OC from validation dataset.

**Figure 8 biomolecules-13-00343-f008:**
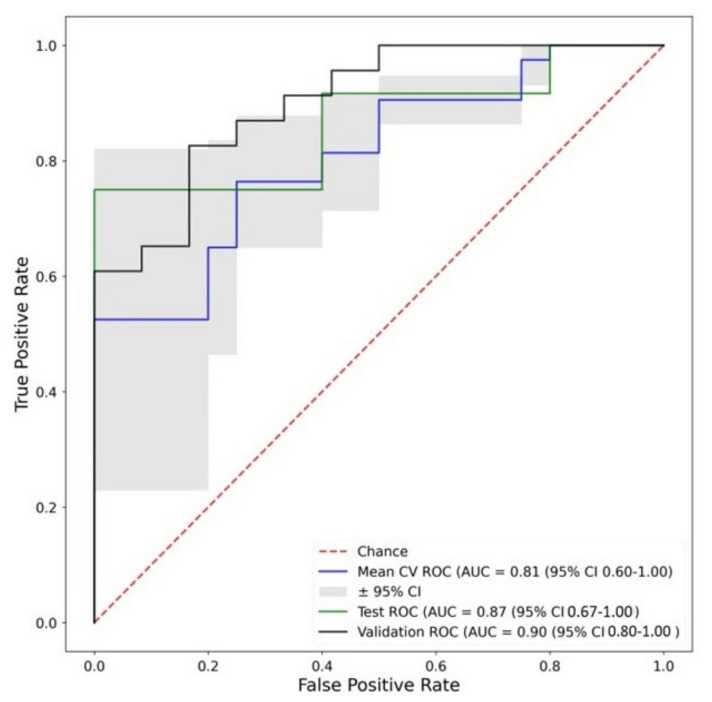
ROC curves of the best performing (by validation AUC) machine learning classifier trained on Fingerprint B—Neural Network. Corresponding Accuracies—Training: 0.78, Test: 0.68, Validation: 0.81. Key: Mean CV ROC—Mean cross validation ROC from training dataset, Test ROC—ROC from test dataset, Validation ROC—ROC from validation dataset.

**Figure 9 biomolecules-13-00343-f009:**
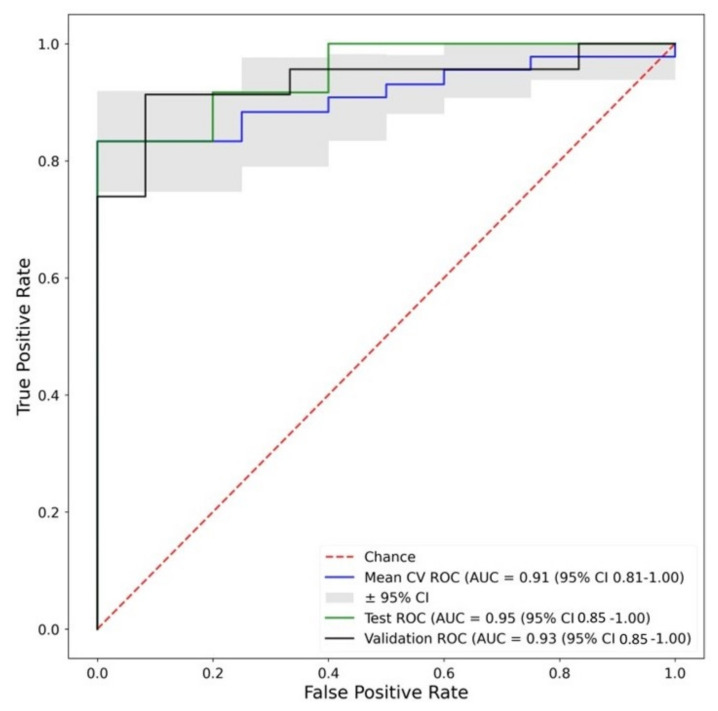
ROC curves of the Random Forest classifier in Fingerprint C. Corresponding Accuracies—Training: 0.79, Test: 0.8, Validation: 0.73. Key: Mean CV ROC—Mean cross validation ROC from training dataset, Test ROC—ROC from test dataset, Validation ROC—ROC from validation dataset.

**Table 1 biomolecules-13-00343-t001:** Distribution of participants across scanners.

Scanner	Training		Test		Validation		Harmonization Batch
	Aortitis	Control	Aortitis	Control	Aortitis	Control	
Discovery 710	14	7	4	4	3	3	1
Gemini TF64	14	11	3	0	0	0	2
Discovery 690	15	3	5	1	9	2	3
Biograph 6 and Biograph 6 True Point	0	0	0	0	5	2	4
Biograph 64 mCT	0	0	0	0	1	2	5
Discovery MI DR	0	0	0	0	6	3	6
Discovery ST and STE	0	0	0	0	0	2	7

Discovery scanners from GE Healthcare—Chicago, IL, USA. Gemini Scanner from Philips Healthcare—Best,
Netherlands. Biograph scanners from Siemens Healthineers—Erlangen, Germany

**Table 2 biomolecules-13-00343-t002:** A description of patient demographics across the three datasets Key—Large vessel vasculitis (LVV), giant cell arteritis (GCA), Takayasu’s arteritis (TAK), Not applicable (n/a), CRP (C-reactive protein), ESR (erythrocyte sedimentation rate).

	Training		Test		Validation	
	**Aortitis**	**Controls**	**Aortitis**	**Controls**	**Aortitis**	**Controls**
Number of Participants	43	21	12	5	19	14
Age at time of scan, years -median (range)	67 (23–85)	67 (41–84)	70 (58–76)	60.5 (49–70)	67 (55–85)	68 (50–79)
Sex (male/female)	11/32	11/10	4/8	2/3	4/15	5/9
LVV type	40 GCA 3 TAK	n/a	12 GCA	n/a	17 GCA 2 TAK	n/a
Prednisolone dose at time of scan, mg -median (range)	0 (0–40)	0 (0–30)	0 (0–40)	0 (0–60)	0 (0–40)	3.5 (0–40)
CRP (mg/L) -median (range)	41 (5–165), not done (n = 8)	n/a	39 (11–149), not done (n = 3)	n/a	36 (10–112), not known (n = 15)	n/a
ESR (mm/Hr) -median (range)	71 (3–143), not done (n = 29)	n/a	37 (n = 1), not done (n = 11)	n/a	90 (12–120), not known (n = 15)	n/a
Blood Glucose (mmol/L) -median (range)	5.5 (4.2–9.9), not known (n = 11)	5.9 (4.6–12), not known (n = 13)	5.8 (5–7.3), not known (n = 3)	5.9 (5.1–7.4), not known (n = 2)	5.8 (4.4–7.5), not known (n = 7)	6.65 (5.4–9.5), not known (n = 2)

**Table 3 biomolecules-13-00343-t003:** Grading of patient dataset based on EANM/SNMMI guidelines [11].

Grade	Training		Test		Validation	
	Aortitis	Control	Aortitis	Control	Aortitis	Control
0	0	21	0	5	0	11
1	1	0	0	0	0	3
2	2	0	0	0	2	0
3	40	0	12	0	17	0
Ground Truth	Grade 3 n = 43	Grade 0 n = 21	Grade 3 n = 12	Grade 0 n = 5	Grade 3 n = 19	Grade 0 n = 14

**Table 4 biomolecules-13-00343-t004:** Features selected for Fingerprint A and C. Key—*SUV (standardized uptake value)*, *GLDM* (*Gray-Level Dependence Matrix*), *GLCM* (*Gray-Level Co-Occurrence Matrix*), *GLRLM* (*Gray-Level Run Length Matrix*), and *GLSZM* (*Gray-Level Size Zone Matrix*).

Top Ten Features Selected in:	
**Fingerprint A**	**Fingerprint C**
GLDM Small Dependence High Gray Level Emphasis	GLRLM Long Run Low Gray Level Emphasis
GLSZM Size Zone Non-Uniformity Normalized	GLSZM High Gray Level Zone Emphasis
GLRLM Gray Level Variance	GLDM Dependence Entropy
GLDM Large Dependence Low Gray Level Emphasis	GLDM Small Dependence High Gray Level Emphasis
GLRLM Long Run Emphasis	GLCM Autocorrelation
GLSZM Gray Level Variance	GLRLM Short Run Emphasis e
First Order Total Energy	GLDM Dependence Non-Uniformity Normalized
GLSZM Large Area Emphasis	First Order Entropy
GLSZM Size Zone Non-Uniformity	GLDM Gray Level Variance
First Order 10-Percentile	GLDM Large Dependence Emphasis

**Table 5 biomolecules-13-00343-t005:** A summary of the diagnostic utility of each explored method.

Qualitative Assessment	Literature AUC 0.81–0.98 [11]					
	Training Accuracy	Training AUC (95% CI)	Testing Accuracy	Testing AUC (95% CI)	Validation Accuracy	Valdiation AUC (95% CI)
SUV Feature -SUV 90th Percentile	0.77	0.91 (0.73–1.00)	0.7	0.93 (0.79–1.00)	0.71	0.85 (0.72–0.99)
Radiomic Feature -GLDM Dependence Entropy	0.55	0.80 (0.61–1.00)	0.7	0.92 (0.74–1.00)	0.60	0.91 (0.82–1.00)
Fingerprint A -Random Forest	0.74	0.90 (0.79–1.00)	0.8	0.98 (0.94–1.00)	0.75	0.92 (0.84–1.00)
Fingerprint B -Neural Net	0.66	0.81 (0.60–1.00)	0.68	0.87 (0.67–1.00)	0.81	0.90 (0.80–1.00)
Fingerprint C -Random Forest	0.79	0.91 (0.81–1.00)	0.8	0.95 (0.85–1.00)	0.73	0.93 (0.85–1.00)

**Table 6 biomolecules-13-00343-t006:** A summary of the diagnostic utility of each explored method when feature harmonization is applied.

Qualitative Assessment	Literature AUC 0.81–0.98 [11]					
	Training Accuracy	Training AUC (95% CI)	Testing Accuracy	Testing AUC (95% CI)	Validation Accuracy	Valdiation AUC (95% CI)
SUV Feature -SUV 90th Percentile	0.69	0.86 (0.66–1.00)	0.7	0.93 (0.79–1.00)	0.67	0.83 (0.68–0.99)
Radiomic Feature—GLDM Small Dependence High Gray Level Emphasis	0.66	0.85 (0.73–0.97)	0.8	0.98 (0.94–1.00)	0.77	0.82 (0.67–0.96)
Fingerprint A—Logistic Regression	0.76	0.86 (0.69–1.00)	0.72	0.90 (0.75–1.00)	0.79	0.93 (0.86–1.00)
Fingerprint B—Neural Net	0.64	0.74 (0.57–0.91)	0.72	0.90 (0.75–1.00)	0.77	0.90 (0.80–1.00)
Fingerprint C—Random Forest	0.81	0.88 (0.72–1.00)	0.62	0.88 (0.71–1.00)	0.7	0.89 (0.79–1.00)

**Table 7 biomolecules-13-00343-t007:** A comparison of key results from different segmentation methods.

	Manual Segmentation Training AUC Mean (95% CI)	Automated Segmentation Training AUC Mean (95% CI)
SUV Feature—SUV 90th Percentile	0.85 (0.77–0.93)	0.86 (0.81–0.91 )
Radiomic Feature—GLSZM High Gray Level Zone Emphasis/GLCM Difference Variance	0.91 (0.84–0.98 )	0.89 (0.87–0.91 )
Fingerprint A—Random Forest	0.91 (0.80–1.0 )	0.85 (0.81–0.89 )
Fingerprint B—Random Forest/Support Vector Machine	0.88 (0.81–0.95 )	0.91 (0.84–0.98 )
Fingerprint C—Random Forest	0.86 (0.78–0.94 )	0.81 (0.74–0.89 )

**Table 8 biomolecules-13-00343-t008:** The diagnostic utility of the explored methods when data with an altered imaging acquisition protocol is added.

	Non-Harmonized		Harmonized	
	**Validation Accuracy**	**Validation AUC (95% CI)**	**Validation Accuracy**	**Validation AUC (95% CI)**
SUV Feature—SUV Mean	0.6	0.72 (0.62–0.83)	0.59	0.72 (0.61–0.82)
Radiomic Feature—First Order Energy/GLDM Dependence Entropy	0.63	0.72 (0.61–0.82)	0.58	0.83 (0.75–0.91)
Fingerprint A—Random Forest/K Nearest Neighbours	0.71	0.80 (0.71–0.89)	0.69	0.72 (0.61–0.82)
Fingerprint B—Perceptron	0.7	0.72 (0.61–0.82)	0.7	0.70 (0.59–0.81)
Fingerprint C—Random Forest	0.48	0.61 (0.50–0.72)	0.6	0.68 (0.57–0.78)

## Data Availability

Code and some data available with reasonable request.

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
