# Peer review of "An Automated Method for Artifical Intelligence Assisted Diagnosis of Active Aortitis Using Radiomic Analysis of FDG PET-CT Images"

_biomolecules, 2023, doi:10.3390/biom13020343_

Round 1

Reviewer 1 Report

The topic of current study was quite interesting and there is a good flow in structure of manuscript. There are some important limitations which clearly discussed by the authors, while they still impact the reached results. I have a few comments:
- The authors used a combination of AI, ML, and CNN in describing the objectives, introduction and even title. The main idea is using CNN algorithm and the authors just simply keep using CNN model in whole text.
- Introduction is written a bit too general. It could be more "to the point". Authors wants to test and validate an already defined CNN mode. That should be the start point of introduction as this is an added analysis to previous study.
- Description of SUV and ROI could be moved to Methods section.
- Due to the importance of the issue to the readers, it could be nice to have Inclusion and Exclusion Criteria once more in Methods section (perhaps very briefly).
 - Figure1 could be revised and be in a smaller size.
- OSEM reconstruction is mostly used within 9 different scanner, which is good to make the received images more similar. However, the authors needs to bring the scanning protocols in more details (good idea to keep as supplementary material).
- Titles of Table 2 and 8 need to be more complete.
- Please add a paragraph in Methods section and explain the statistical analysis plans. Then, the readers could expect what results/outcomes come next.
-Placing the Tables and Figures should be totally updated. They should insert in the text, close to the paragraph they used.
- Looking at the literature, we can find a few studies aortic wall segmentation of PET/CT scans, which is mentioned as a limitation by authors. I think the different approaches of other studies with
aortic wall segmentation could be compared in Discussion section:
https://pubmed.ncbi.nlm.nih.gov/34291373/
https://pubmed.ncbi.nlm.nih.gov/33982202/

- Discussion section is a bit long, but OK. Perhaps the authors could add a few sub-headings to the discussion as well in order to be easier to the readers to follow the different comparisons, as we already have several results. 
- Conclusion needs to be updated. Bring the summary of most important findings.

Author Response

We would like to thank you for your insightful comments regarding our paper. We believe the suggested changes have improved the quality and brought to light flaws we had missed.

Does the introduction provide sufficient background and include all relevant references? : Must be improved

The changes made to the introduction have been addressed more completely in the response to the written comment below.

Is the research design appropriate?: can be improved

The description of the research design has been improved and any limitations that shaped our research design have been more clearly described.

Are the results clearly presented? : Must be improved

The figure and tables have been better placed as described in a response below and some less prioritised results have been removed. Additionally, I hope the steps of a radiomic workflow being added at the star helps clarify the results corresponding to each step.

Are the conclusions supported by the results? : Must be improved

The conclusion has been rewritten to include key results and their relationship to the conclusions drawn.

The authors used a combination of AI, ML, and CNN in describing the objectives, introduction and even title. The main idea is using CNN algorithm and the authors just simply keep using CNN model in whole text.

A combination of simple ML classifiers for diagnosis and CNN for segmentation were used so the terms were used where relevant and AI used in general descriptions. Before resubmission we have proofread the manuscript to ensure the correct terms were always used, the acronyms explained at first use, and the methods made clearer.

Introduction is written a bit too general. It could be more "to the point". Authors wants to test and validate an already defined CNN mode. That should be the start point of introduction as this is an added analysis to previous study.

The introduction has been made more concise by introducing only essential background information, important information about the previous paper and the key aims of this paper.

Description of SUV and ROI could be moved to Methods section.

These definitions were moved to the methods section as advised.

Due to the importance of the issue to the readers, it could be nice to have Inclusion and Exclusion Criteria once more in Methods section (perhaps very briefly).

The inclusion and exclusion criteria have now been described in more detail in the methods section.

Figure1 could be revised and be in a smaller size.

The figure has been redesigned and made to a more appropriate size.

OSEM reconstruction is mostly used within 9 different scanner, which is good to make the received images more similar. However, the authors needs to bring the scanning protocols in more details (good idea to keep as supplementary material).

More details have been added regarding the imaging protocol in both the methods section and supplemental material.

Titles of Table 2 and 8 need to be more complete.

The titles of both tables have been elaborated.

Please add a paragraph in Methods section and explain the statistical analysis plans. Then, the readers could expect what results/outcomes come next.

A paragraph has been added to the end of the methods section where all statistical analysis is discussed in one place.

Placing the Tables and Figures should be totally updated. They should insert in the text, close to the paragraph they used.

A new figure and table placement code has now been added to our latex file to ensure all figures remain where we placed them in the text.

Looking at the literature, we can find a few studies aortic wall segmentation of PET/CT scans, which is mentioned as a limitation by authors. I think the different approaches of other studies with aortic wall segmentation could be compared in Discussion section: https://pubmed.ncbi.nlm.nih.gov/33982202/   https://pubmed.ncbi.nlm.nih.gov/34291373/

The studies you have provided plus a few others have been discussed and added to the discussion section. In summary most others available require contrast enhancement which was not an option to us. The example studies provided use a predefined thickness to approximate as a wall. We considered this approach but decided the limitations outweighed the benefits.

Discussion section is a bit long, but OK. Perhaps the authors could add a few sub-headings to the discussion as well in order to be easier to the readers to follow the different comparisons, as we already have several results.

The discussion section has been divided into subsections and titled appropriately to make the main discussion points clearer and easier to follow.

Conclusion needs to be updated. Bring the summary of most important findings.

The conclusion has been rewritten to include key results and their relationship to the conclusions drawn.

Reviewer 2 Report

This paper studies the approach for AI assisted diagnosis of active aortitis. The followings are some comments.

l  This paper is a follow-up work of ref. [30]. The authors may want to clearly indicate what is new in the present manuscript when compared with previous work. It is the authors’ responsibility to convince me (one of the reviewers) that the present manuscript contains sufficient new materials to be considered as a new publication, not a duplicate publication.

l  The results on Table 4 seems unreasonable to me. For example, the perceptron classifier has the validation accuracy of merely 0.5. Yet, it has an AUC of 0.89, even higher than that of the logistic regression (which has accuracy of 0.79). As the diagnosis of active aortitis is a binary classification problem, the accuracy of 0.5 is really nothing more than tossing a coin. I understand that the number of aortitis patients and control patients are different, and therefore some balance work was carried out. Still, in a typical case, the AUC is, in a rough sense, proportional to accuracy. The authors are suggested to study the meaning of AUC, and then carefully examine what causes this (apparently) unreasonable phenomenon.

l  As there are several findings presented in the manuscript, it would be easier for readers to follow if the authors can provide a subsection to describe how an automated clinical decision tool could be built, preferably with a figure showing the processing flows.

l  The followings are some minor points to correct.

                         i.              Line 114 of page 4 has ??. Please provide the DSC equation.

                       ii.              Line 230 of page 8 has ??

                      iii.              Spell out LR on line 209 of page 8, as LR could be logistic regression or linear regression (or others).

                      iv.              The stochastic gradient descent (on line 210 of page 8) is an optimization approach, but not the formal name of any kind of classifier. I suppose that authors used linear SVM with SGD optimizer in the experiments.

Author Response

We would like to thank you for your insightful comments regarding our paper. We believe the suggested changes have improved the quality and brought to light flaws we had missed.

Are the results clearly presented? : Must be improved

The description of results has been clarified and some results with a lower priority have been moved to supplemental material. Additionally I hope the steps of a radiomic workflow being added at the start helps clarify the results corresponding to each step.

Are the conclusions supported by the results? : Can be improved

The conclusion has been rewritten to include key results and explain their relationship to the conclusions drawn.

 This paper is a follow-up work of ref. [30]. The authors may want to clearly indicate what is new in the present manuscript when compared with previous work. It is the authors’ responsibility to convince me (one of the reviewers) that the present manuscript contains sufficient new materials to be considered as a new publication, not a duplicate publication.

I have now added to the introduction an explicit description of the progress made in this paper compared to our previous paper. In summary the key differences are adding an automated segmentation method that still maintains a similar level of diagnostic performance, and secondly validating the results of the initial paper in a larger multiple centre data set to confirm transferability. In both cases this takes our initial proof of concept and displays its ability to be used in a wider clinical scenario.

The results on Table 4 seems unreasonable to me. For example, the perceptron classifier has the validation accuracy of merely 0.5. Yet, it has an AUC of 0.89, even higher than that of the logistic regression (which has accuracy of 0.79). As the diagnosis of active aortitis is a binary classification problem, the accuracy of 0.5 is really nothing more than tossing a coin. I understand that the number of aortitis patients and control patients are different, and therefore some balance work was carried out. Still, in a typical case, the AUC is, in a rough sense, proportional to accuracy. The authors are suggested to study the meaning of AUC, and then carefully examine what causes this (apparently) unreasonable phenomenon.

We believe this discrepancy comes from a combination of using balanced accuracy and the classifier being bias towards diagnosing positive cases. When looking at the results for this that specific fingerprint all cases classified as positive. This bias is known to increase AUC. As FP are better than FN this bias is not always a bad thing but is why we use balanced accuracy to check the bias is not to this extent. As the accuracy is weighed to deal with the class imbalance it makes the difference in the two metrics appear larger. The original accuracy measurement was approximately 70% due to us having more aortitis cases than controls. It also must be considered that the AUC is based on probabilities whereas accuracy is based on predictions. We chose to present the result of accuracy = 0.5 as no classifiers did well for that fingerprint and this was technically the best even if it is poor. By presenting this we can show this method is not effective.

As there are several findings presented in the manuscript, it would be easier for readers to follow if the authors can provide a subsection to describe how an automated clinical decision tool could be built, preferably with a figure showing the processing flows.

The results have been edited to make following them clearer. Following your suggestion, we added an introduction to the methods to clarify each step in making a radiomic model. As much as possible we attempted to keep headings matching these steps so the methods and results could be more easily followed.

 Line 114 of page 4 has ??. Please provide the DSC equation.

The equation has been added and the reference to it fixed.

 Line 230 of page 8 has ??

We believe this error has now been fixed but appears to keep reoccurring with this specific figure. If it still appears unresolved, we will remove this figure.

 Spell out LR on line 209 of page 8, as LR could be logistic regression or linear regression (or others).

LR has now been written out fully.

The stochastic gradient descent (on line 210 of page 8) is an optimization approach, but not the formal name of any kind of classifier. I suppose that authors used linear SVM with SGD optimizer in the experiments.

I have added an explanation to the manuscript as well but this result refers to the scikit learn SGD classifier that tries several different linear classifiers with SDG optimization. The specific linear classifier can be different every time as it is treated like another hyperparameter to optimize.

Round 2

Reviewer 1 Report

All the mentioned points have been covered and discussed in updated version and, therefore, the manuscript could be accepted after minor language changes.

Author Response

Thank you for your review. I will proof read before submission to ensure the language corrections are made. 

Reviewer 2 Report

Dear authors,

Thanks for revising the manuscript. I have just only one concern left, which is related to accuracy and AUC. I raised this concern in the first-run comments. Firstly, let me briefly explain the meaning of AUC before I continue. The AUC computes the area under the ROC curve. If a binary classifier has only one output value, then we can assign the incoming sample to class A if the predicted output value is greater than the threshold; otherwise, class B. Conceptually, adjusting the threshold value changes the accuracy of any classifier. To fairly compare the performance of different classifiers (which may have different threshold values), the ROC curve was introduced to show the change of the true positive rate against fault positive rate over different threshold values (of one approach). Thus, by reading the ROC curves, we are able to have a global picture about the prediction performance of a particular classifier. The AUC was then introduced to use one value to replace the entire ROC curve.

Knowing the basis of AUC, I think the authors can at least do the followings to convince me that the experimental results are correct (such as the numbers in fingerprint B in Table 6):

l   Check the number of output values in the classifier (one value or two values).

l   If the classifier has one value, figure out how the class of a sample is assigned (usually by assuming a threshold value of 0.5). The accuracy is computed based on the assigned class of each sample.

l   Change the threshold value to see the variation of accuracy along with the threshold. Find the highest accuracy value. This value should be much greater than 0.5. Show the curve in the revised version with discussions. Using one classifier as an example is enough.

l   If the classifier has two values, one value for one class, the incoming sample is usually assigned to the class with a larger output value. In this case, carefully explain how to apply a threshold to these two output values to obtain the AUC. Also, change the threshold value to have a curve of accuracy against threshold value. Show the curve and discuss.

I understand the authors are not expert in machine learning areas. However, to be responsible for a published paper, the authors must make sure that everything is correct. Therefore, conducting a verification procedure to double check the results is necessary, especially when some values in the experimental results (tables) seem to be counter-intuitive.
